# Cycles of protein condensation and discharge in nuclear organelles studied by fluorescence lifetime imaging

Artem Pliss[1], Svitlana M. Levchenko [2], Lixin Liu [3], Xiao Peng[2], Tymish Y. Ohulchanskyy[1,2], Indrajit Roy[4], Andrey N. Kuzmin [1], Junle Qu[2] & Paras N. Prasad[1,2]

Nuclear organelles are viscous droplets, created by concentration-dependent condensation and liquid–liquid phase separation of soluble proteins. Nuclear organelles have been actively investigated for their role in cellular regulation and disease. However, these studies are highly challenging to perform in live cells, and therefore, their physico-chemical properties are still poorly understood. In this study, we describe a fluorescence lifetime imaging approach for real-time monitoring of protein condensation in nuclear organelles of live cultured cells. This approach unravels surprisingly large cyclic changes in concentration of proteins in major nuclear organelles including nucleoli, nuclear speckles, Cajal bodies, as well as in the clusters of heterochromatin. Remarkably, protein concentration changes are synchronous for different organelles of the same cells. We propose a molecular mechanism responsible for synchronous accumulations of proteins in the nuclear organelles. This mechanism can serve for general regulation of cellular metabolism and contribute to coordination of gene expression.

[1] Institute for Lasers, Photonics and Biophotonics and Department of Chemistry, State University of New York, Buffalo, NY 14260, USA. [2] Key Laboratory of Optoelectronic Devices and Systems of Ministry of Education and Guangdong Province, College of Optoelectronic Engineering, Shenzhen University, Shenzhen 518060, China. [3] School of Physics and Optoelectronic Engineering, Xidian University, Xi'an, Shaanxi 710071, China. [4] Department of Chemistry, University of Delhi, Delhi 110007, India. These authors contributed equally: Artem Pliss, Svitlana M. Levchenko. Correspondence and requests for materials should be addressed to J.Q. (email: jlqu@szu.edu.cn) or to P.N.P. (email: pnprasad@buffalo.edu)

A spectacular feature in the architecture of the cell nucleus is that thousands of soluble proteins self-associate and spatially segregate into several membrane-less nuclear organelles, which play a fundamental role in cellular regulation. Most notable examples of nuclear organelles include nucleoli, which accommodate the synthesis of ribosomal RNA (rRNA), nuclear speckles that store proteins involved in messenger RNA (mRNA) synthesis and processing, and the Cajal bodies, which catalyze the biogenesis of small ribonucleoproteins for pre-mRNA splicing[1]. In addition, proteins condensation is essential for compaction of inactive genomic sequences. Large heterochromatin domains are formed through the transient interactions of DNA-binding proteins[2].

Although the biochemical composition and functions of nuclear organelles have been thoroughly investigated over the last decades, the physico-chemical mechanisms behind their formation have just begun to emerge. Proteins, which tend to populate the nuclear compartments, are enriched in hydrophobic as well as charged amino acids. In the highly crowded environment, free interactions between the hydrophobic, non-polar and charged motifs lead to self-association, condensation and segregation of miscible types of proteins into discrete spherical droplets, measuring up to several micrometers, via the liquid–liquid phase separation mechanism[3,4]. It can be assumed that a large difference between the dielectric properties of the protein condensates and nucleoplasm[5] is one of the factors driving the formation and maintenance of the nuclear organelles.

Formation of the nuclear organelles has been extensively investigated in relation to their function[6]. It has been postulated that the elevated concentration of interacting macromolecules facilitates biochemical processes in organelles[4]. Moreover, an increase in the molecular concentration is additionally amplified through the macromolecular crowding phenomenon. One major effect of macromolecular crowding is that accumulation of proteins reduces the volume available for diffusion, which increases the rates of biochemical reactions by orders of magnitude[7–10]. At the same time, the phase separation does not produce the truly isolated environments, and the macromolecular content of nuclear organelles is actively exchanged with the nucleoplasm. The residence time of fluorescence-labeled proteins into different nuclear organelles ranges from single seconds to several tens of seconds[11].

A growing body of evidence shows that the condensation of proteins correlates with the cellular physiologic states. The morphology and molecular composition of nuclear organelles is compromised by degenerative and metabolic diseases, as well as by cancer[12,13]. It has been asserted that understanding of mechanistic principles that underlie their assembly and disassembly may provide key to diagnostics of abnormal cellular conditions, and help to develop new treatment approaches[10,14]. However, such studies are exceedingly challenging, since proteomes of membrane-less organelles contain up to several thousand types of proteins, which could not be simultaneously monitored[15].

To investigate the physico-chemical properties of the nuclear organelles, the quantitative imaging tools, have been developed[8,16]. In one pioneering study, several nuclear organelles were isolated from amphibian oocytes and their refractive indexes (RIs) were measured by interferometry microscopy. Next, obtained RIs were used to estimate the protein concentrations into the single organelles, for the first time[17]. However, this technique involves cell disruption and, inherently, is not compatible with physiological conditions.

Until now, kinetics of proteins condensation were mostly investigated in solutions containing isolated nuclear proteins, RNA as well as inert macromolecules added for simulation of the intercellular molecular crowding[9,18–20]. Therefore, it is not quite clear, to what extent any experimental observations obtained in artificial systems, reflect molecular behavior in live cells[21].

An alternative solution to this long-standing problem could be found using the non-invasive fluorescence lifetime imaging (FLIM) technique. FLIM approach involves fluorescent labeling of the nuclear organelles and exploits the inverse quadratic correlation between the fluorescence lifetimes of fluorophores and RI in their immediate microenvironment. This dependence is defined by Strickler-Berg equation[22–25] and is detailed in Methods. Previously, the inverse correlation between fluorescence lifetime and local RI was validated for solutions of fluorescent proteins such as GFP[25,26]. More recently, using GFP and tdTomato fusions, we reported on the sensitivity of FLIM to RI changes in live cells throughout the cell cycle progression or drug–cell interactions[27,28].

After decades of research, RI is determined as the only factor, changes in which clearly correlate with changes in the fluorescence lifetime of fluorescent proteins (in the absence of FRET)[27,29–31]. At the same time, variations of pH do not change the fluorescence lifetime of GFP-like proteins. While the absorption spectra and emission intensity of GFP are pH dependent, fluorescence lifetime remains unchanged over a broad pH range, when the excitation is performed under ~480 nm[32]. Similarly, variations in viscosity affect the anisotropy of fluorescent proteins, but not their lifetime[23]. This photochemical stability is attributed to the common structure of fluorescent proteins, which involves the internal location of fluorophores within barrel-shaped domains, which protect them from interactions with the surrounding medium[33]. At the same time, the sensitivity of fluorescence lifetime decay to local RI offers an opportunity to use fluorescent proteins as probe for monitoring the proteins concentration in their immediate environment.

In this study, we employ FLIM to investigate the cornerstone principle of nuclear organelles formation—a concentration-dependent condensation of nucleoplasmic proteins. We probe the changes in protein concentrations in major nuclear organelles participating in RNA synthesis and processing—nucleoli, nuclear speckles, Cajal bodies as well as in transcriptionally dormant heterochromatin domains. Our data indicate that the balance between the rates at which proteins coalesce and exit back to the nucleoplasm is not constant. Instead, we detect large cyclic changes in the protein concentration in the studied nuclear organelles. Moreover, we find a significant level of correlation between the rates of proteins condensation to different nuclear organelles of the same cell. We suggest that reported here cyclic changes in protein condensation facilitate a coordination between nucleolar and nucleoplasmic RNA synthesis.

## Results

**Calibration of FLIM set-up.** As discussed above, the fluorescence lifetime of a fluorophore is linked to the RI of its microenvironment through an inverse quadratic relation, described by Strickler-Berg equation[22–25]. We applied this dependence for quantitative monitoring of proteins concentration changes in fluorescence-labeled organelles.

In the first experiments, we calibrated our FLIM set-up for detection of variations in the protein concentrations. Recombinant tdTomato Fluorescent Protein and Enhanced Green Fluorescent Protein (EGFP) were mixed with solutions containing different amounts of bovine serum albumin (BSA) and their fluorescent lifetimes were measured. The obtained data indicate a linear inverse dependence of EGFP and tdTomato fluorescence lifetimes on the concentration of BSA, in agreement with the previous studies[24,25,27,28]. For EGFP the fluorescence lifetime in phosphate

buffered saline (PBS) was 2.98 ns, and reduced to 2.96, 2.91, 2.89, and 2.83 ns in solutions of 50, 100, 150, and 200 mg/mL of BSA in PBS, correspondingly. Similarly, for tdTomato, we observed a decrease in the fluorescence lifetime from ~3.16 ns in PBS to 3.13, 3.08, 3.05, and 3.00 ns, with the corresponding increase in BSA concentrations (Fig. 1a, b).

Next, we calibrated the response of fluorescent protein fusions lifetimes to the changing concentrations of soluble proteins in their subcellular environment. In these experiments, cells were transfected with either Fibrillarin-tdTomato, ASF/SF2-EGFP, or Coilin-EGFP to label nucleoli, nuclear speckles and Cajal bodies, respectively[11,34,35], and then fixed, permeabilized and immersed in the series of BSA/PBS solutions. Our calibration experiments contained also cells transfected with histone H2B–EGFP[11], as an additional control. Histone H2B is immobilized in the nuclear chromatin, which prevents potential aggregation of fluorescent proteins during the fixation, and facilitates exposure to changing concentrations of BSA.

In the fixed cells, Coilin-EGFP, ASF/SF2-EGFP, and histone H2B-EGFP fluorescent proteins exhibited fluorescence lifetime averaging at 2.31, 2.40, and 2.42 ns, respectively, and Fibrillarin-tdTomato at ~3.05 ns. These fluorescence lifetime values of all protein fusions were significantly shorter as compared to free EGFP and tdTomato. However, similar to experiments above, we recorded a substantial shortening of the fluorescent lifetimes when the cells were immersed into BSA/PBS solutions. All tested fluorescence fusions, demonstrated nearly an identical response to the changing concentrations of BSA, as demonstrated by close values of their dependence slopes (Fig. 1c). At 150 mg/mL of BSA (the highest tested protein concentration), we recorded ~100 ps reduction in the fluorescence lifetimes for all tested proteins as compared to PBS.

Dependence of EGFP and tdTomato fluorescence lifetimes on the concentrations of BSA in their microenvironment was further used for monitoring protein concentration changes in the nuclear organelles. It should be noted that measurements of protein concentrations in a cell could be distorted in the proximity of cellular membranes, which have a high RI and induce the shortening of the lifetime of fluorophore. At the same time, the calibration charts presented in Fig. 1a, b allow for quantitative monitoring of the changes in the protein concentrations in single nuclear organelles of the same cell.

To characterize the sensitivity and reproducibility of FLIM technique for monitoring of nuclear organelles we performed control experiments. Cells expressing ASF/SF2- tdTomato were fixed with 2% formaldehyde in PBS, placed in a stage-mounted incubator at 37 °C, and fluorescence lifetimes images were acquired every 10 min. These measurements defined the experimental error of our set-up, wherein the difference between florescence lifetimes values at different time points was within a ~7 ps margin (Supplementary Fig. 1). In another series of control experiments, the cells were transfected with histone H2B-EGFP, which is an immobile protein and does not accumulate in the nuclear organelles. Here, the fluorescence lifetime data were acquired in live cells, which were maintained in the incubator throughout the imaging session, same as in experiments above. We found that while the fluorescence lifetime of H2B-EGFP could change above the experimental error, the measurements were relatively reproducible. The amplitude of fluorescence lifetime fluctuations was usually below 15 ps within one hour of observation, indicating moderate fluctuations of the RI in the nucleoplasm of live cells (Supplementary Fig. 2).

**Proteins condensation occurs in cycles**. Nuclear organelles were visualized by using tdTomato and EGFP fluorescence fusions[36]. In these experiments, cells were transfected by either ASF/SF2-EGFP, Fibrillarin-tdTomato, Coilin-EGFP or Heterochromatin protein 1 beta (HP1β-EGFP) DNA constructs[11,34]. Next day after transfection, cells were expressing these fluorescent proteins, and the signal was accumulated in the nuclear speckles, nucleoli, Cajal bodies, and the heterochromatin domains, respectively[11,35].

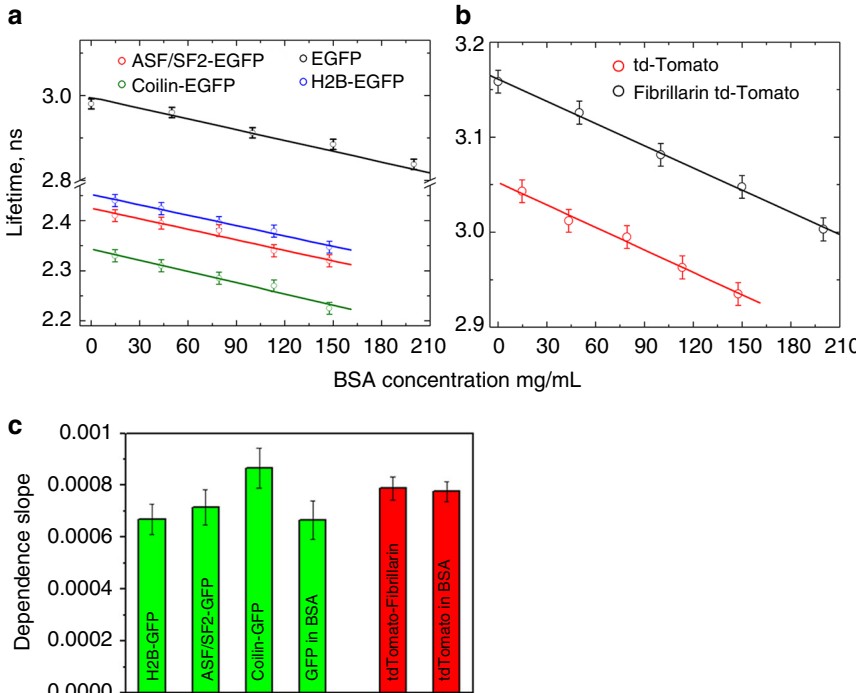

**Fig. 1** Calibration of fluorescent proteins lifetimes in bovine serum albumin solutions. **a**, **b** Dependence of fluorescence lifetimes of EGFP and tdTomato on the concentration of bovine serum albumin (BSA) in solution. **c** Dependence slopes calculated for plots **a** and **b**. Error bars are St. Dev. ($n = 15$ for each calibration point)

During the imaging, cells were maintained under the physiological conditions at 37 °C. Fluorescence lifetime images were sequentially acquired every 10 min from the cells with representative morphology. Studied organelles were manually segmented, and the fluorescence lifetime values for the same type of organelles in a cell were averaged for each time point. Data for each type of the studied organelles are presented below.

**Nuclear speckles.** Nuclear speckles were visualized by the signal from ASF/SF2-EGFP. As commonly observed with this construct, it was present both in nucleoplasm as well as in multiple, interconnected domains[11]. The shape of the nuclear speckles did not visibly change over the imaging time and the fluorescence intensity was relatively uniform. In contrast, mapping of the fluorescence lifetime values indicated that the condensation of proteins in the nuclear speckles is variable, producing densely packed protein-rich pockets (Fig. 2a; orange-to-red color) as well as loose domains with relatively low protein concentrations, shown in blue on the FLIM images (Fig. 2a).

As we discussed above, an intriguing property of the phase-separation in the cell nucleus is that proteins have only transient associations with the nuclear organelles. It was previously demonstrated that proteins rapidly shuttle between the nuclear organelles and nucleoplasm[11,34]. Thus, a quantitative model describing formation and maintenance of the nuclear organelles, has to address two opposite processes: the condensation of proteins into droplets and their release back to the nucleoplasm. However, it is not clear whether these processes are maintained in a steady equilibrium, or whether such a balance is compatible with the stochastic cellular environment.

To understand how the interplay of proteins condensation and release change the molecular environment in the nuclear speckles,

the cells expressing ASF/SF2-EGFP construct were monitored in time. In these experiments, the FLIM images were acquired with 10 min intervals, nuclear speckles were segmented and the fluorescence lifetime values were quantified and averaged in all speckles for each studied cell. Using fluorescence lifetime of ASF/SF2-EGFP as a real-time sensor of absolute concentrations of proteins in accordance with the calibration chart (Fig. 1a), we recorded, in most cells, concentration fluctuations in the range of ~10–40 mg/mL between the subsequent time points (Fig. 2b, c). The concentration changes were less significant when the cells were kept at the room temperature, apparently due to lower diffusion of intracellular proteins (Supplementary Fig. 3).

It has been well established, that the nuclear speckles do not accommodate RNA synthesis. Instead, they serve as a reservoir for storage of various components of mRNA synthesis machinery, including RNA polymerase II, transcription factors and splicing proteins[37]. Thus, the cyclic changes in the concentration of proteins (Fig. 2b, c) are likely produced by net redistribution of these proteins between the nuclear speckles and the sites of pre-mRNA synthesis and processing, located outside of these nuclear organelles.

**Nucleolus.** For these series of experiments, nucleoli were visualized via Fibrillarin-tdTomato fusion protein. Each studied nucleolus contained focal sites with high-concentrations of proteins, typically in the inner regions of these nuclear organelles. It is known, that the condensation of nucleolar proteins creates internal compartments within the nucleoli, including the fibrillar centers (FC), dense fibrillar components (DFC), and granular components. The accumulations of fibrillarin typically overlap with active ribosomal genes in the DFCs, as discovered by correlative optical and electron microscopy[38,39]. In addition,

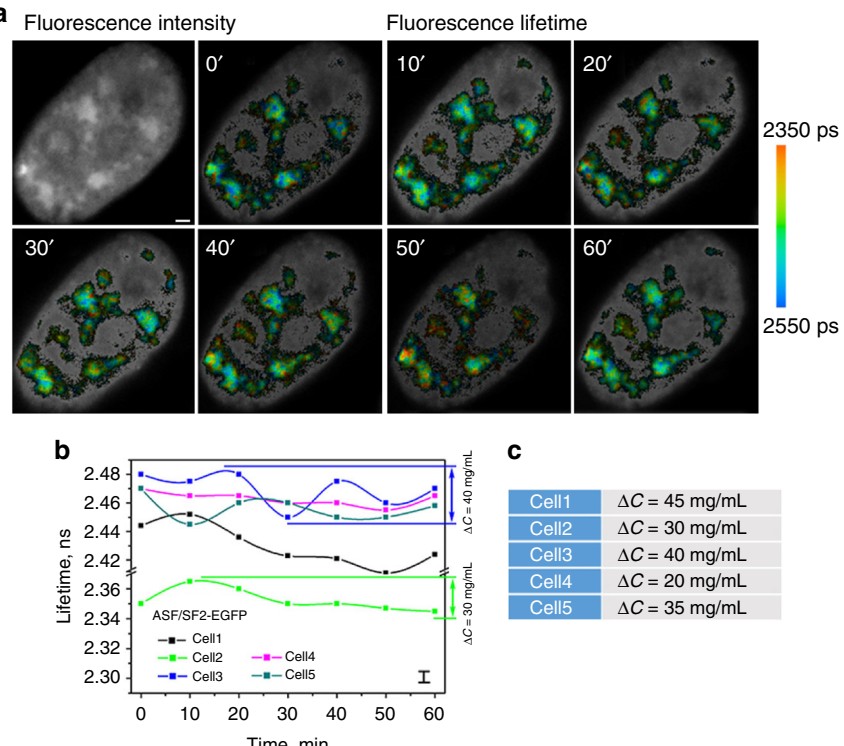

**Fig. 2** Cyclic changes in absolute concentrations of proteins in nuclear speckles. **a** Fluorescence intensity image (upper left) and fluorescence lifetime images, of a live cultured cell expressing ASF/SF2-EGFP. Images were acquired with 10 min intervals. Scale bar is 1 μm. **b** Fluorescence lifetime changes in the nuclear speckles of single cells. Corresponding changes in absolute protein concentrations in the nuclear speckles for two representative cells are shown on the right y-axis. Error bar in the bottom right corner shows the instrument measurement error. **c** Protein concentration changes (ΔC) per same cell

fibrillarin is present in the FCs[40]. FLIM data indicate that the variations in absolute protein concentrations in the individual DFCs and FCs produce corresponding differences in local RI, and result into irregular lifetime pattern. By averaging data from all nucleoli in the cell, we observed significant fluctuations of protein concentration values, similar to changes observed in the nuclear speckles (Fig. 3a).

The amplitude of protein concentration fluctuations was close to that in the nuclear speckles, demonstrating changes in the range from ~30 to ~100 mg/mL, within 10 min intervals (Fig. 3b, c). In addition, high magnification FLIM images suggest that the accumulation pattern of proteins for different nucleolar sub-compartments may change in time as well (Supplementary Fig. 4). At the same time, Fibrillarin-tdTomato lifetime changes in different sub-nucleolar domains were typically in unison to the averaged for nucleolus changes. We also observed that the lifetime fluctuations in the different nucleoli of the same cell were synchronized (Supplementary Fig. 5).

**Cajal nuclear bodies**. In parallel, we investigated the fluorescence lifetime changes in Cajal nuclear bodies. These organelles, which participate in biogenesis of pre-mRNA splicing factors, represent sub-micrometer RNA and protein-rich droplets[41]. In HeLa cells expressing coilin-EGFP, a marker protein for this nuclear organelle, about ten bright foci could be typically counted. We monitored the fluorescence lifetime of this protein fusion, same as in the experiments above. Our data show that protein

concentrations in Cajal nuclear bodies undergo cyclical changes similar to that in nuclear speckles (Supplementary Fig. 6).

**Heterochromatin domains**. Heterochromatin domains in eukaryotic cells accumulate tens of proteins involved in DNA compaction and epigenetic silencing[42]. Heterochromatin domains were visualized using EGFP fusion of HP1 β, which is known for dynamic shuttling between heterochromatin domains and pool of nucleoplasmic proteins[34]. Cells transfected with H1β-EGFP demonstrated several irregularly shaped dense domains consistent with the distribution pattern of this protein in HeLa cells[11].

Similar to measurements performed in other nuclear organelles, FLIM indicates periodic changes in the proteins condensation levels. We found the concentration fluctuations up to 70 mg/mL within 60 min observation periods (Supplementary Fig. 7).

**Synchrony in proteins condensation in different organelles**. It has been long postulated that despite the spatial separation, multiple biochemical processes in different nuclear organelles are coordinated in time[43]. However, the mechanisms for such coordination are largely unknown. For instance, if the activity of the nuclear organelles is dependent on the levels of proteins accumulation by the concentration-dependent condensation, then the question arises as to how these processes can be coordinated for different organelles.

To gain an insight into co-regulation of major metabolic activities in the nuclear organelles, we monitored the changes in

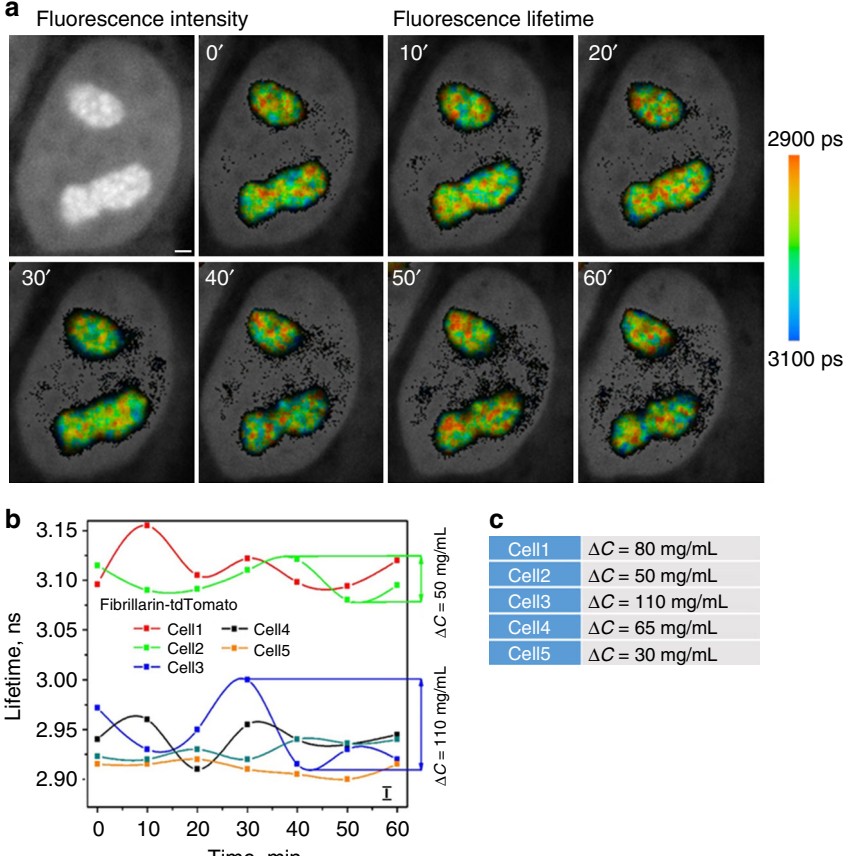

**Fig. 3** Cyclic changes in absolute concentrations of nucleolar proteins. **a** Upper left panel represents fluorescence intensity images of Fibrillarin-tdTomato in live cultured cell, other panels demonstrate fluorescence lifetime data with 10 min intervals. Scale bar is 1 μm. **b** Averaged fluorescence lifetimes of tdTomato in nucleoli for five cells are plotted over the time. Error bar in the bottom right corner shows the instrument measurement error. **c** ΔC demonstrates the amplitude of the protein concentration changes per cell as shown in charts on **b**

the protein content in the nucleoli and the nuclear speckles in the same cells. For these studies, the cells were simultaneously transfected with Fibrillarin-tdTomato and ASF/SF2-EGFP and both fluorescence signals were sequentially collected in the same cell imaging series. The fluorescence lifetimes values were averaged for the whole organelle and changes in the protein concentrations between each time point were calculated.

We found that the concentration changes in these different organelles of the same cell were frequently synchronized (Fig. 4). To evaluate correlation between the changes of protein concentration in the nucleoli and the nuclear speckles of the same cell, we utilized Pearson coefficients ($r$). The Pearson coefficient characterizes the strength and direction of a linear relationship between two variables on a [−1 to 1] scale, where 0 corresponds to the absence of any correlation, −1 total negative and 1 total positive correlation. The Pearson coefficients were found in a positive range, varying between ~0.3 and 0.6 for different cells (Fig. 4b). We thus concluded that the cycles of protein accumulation and release in the nuclear speckles and the nucleoli of the same cell tend to correlate with one another.

In another series of experiments, we comparatively analyzed fluctuations of protein concentrations in the H1-β EGFP stained heterochromatin domains and nucleoli stained with tdTomato (Supplementary Fig. 8). Similar to experiments above, we report a striking correlation between two processes. The Pearson coefficients between the rates of protein condensation were even more positive, ranging between ~0.3 and 0.9 for different cells (Supplementary Fig. 8). Thus, our data indicate significant synchrony in condensation of proteins between different nuclear organelles of the same cells.

## Discussion

The compelling theory that the nuclear organelles are formed through random electrostatic and hydrophobic interactions of soluble proteins, which cause their condensation and phase-separation from the nucleoplasm, rather than through specific molecular interactions, has increasingly gained support[4,44,45]. Consistent with it, proteins of the nuclear organelles contain both charged and neutral motifs, which facilitate their self-assembly into miscible fractions. Condensation and phase-separation of proteins are further promoted because of their generally non-polar properties. The dielectric constant ($\varepsilon$) of proteins is ranging from 6–7 in their internal domains, to 20–30 on the surface, which is far lower than that of water, around 80 at room temperature[5,44,46]. Such a significant difference between the $\varepsilon$ of proteins and $\varepsilon$ of water makes condensation of soluble proteins energetically beneficial; this can be one of the factors driving the formation of the nuclear organelles.

On the other side, it is evident that an increase in the local concentration of proteins, resulting from their condensation, creates a concentration gradient, which is accompanied by the opposite process of diffusion. The equilibrium between proteins net condensation and net diffusion-out is reached at a minimum potential energy of the system, where the phase separation process occurs. This equilibrium can define the concentration of proteins in the nuclear organelles.

However, the experimental verification of these complex macromolecular processes in live cultured cells has lagged behind. Conventional techniques typically involve cell destruction and cannot probe the levels of protein condensation in intact nuclear organelles. Bound by these technical limitations, the intriguing

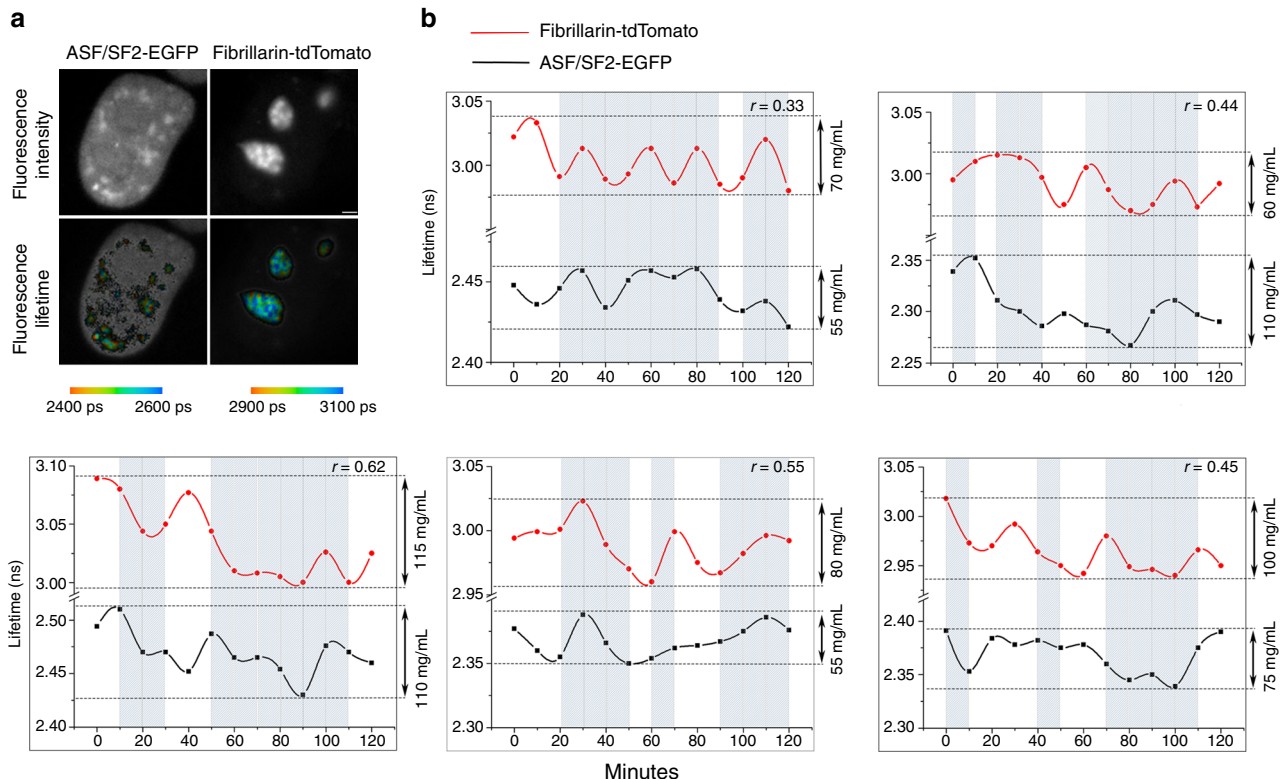

**Fig. 4** Correlation of protein concentration changes in nucleoli and nuclear speckles. Cells expressing Fibrillarin-tdTomato and ASF/SF2–EGFP were used for simultaneous fluorescence lifetime imaging (FLIM) imaging of the nucleoli and the nuclear speckles. **a** Fluorescence intensity and fluorescence lifetime images of ASF/SF2-EGFP and tdTomato in the same cell. Scale bar is 1 μm. **b** Fluorescence lifetimes for Fibrillarin-tdTomato and ASF/SF2-EGFP were monitored for same cells and shown on the left $y$-axis. Corresponding calculated ranges of protein concentrations are shown on the right $y$-axis. Synchronized for both organelles changes in lifetime/concentration are marked by shaded boxes. Pearson correlation coefficients ($r$) between the fluctuations of the protein concentrations in both organelles for each cellular data set are 0.33, 0.44, 0.62, 0.55, and 0.45, as indicated

role of protein condensation into cellular regulation remains virtually unexplored. It has not been defined whether the protein concentrations in the nuclear organelles are stable or dynamically change over time in response to the cell growth, differentiation, malignant transformations or drug–cell interactions. Our FLIM approach provides non-invasive means for such studies in live cell systems.

Moreover, the fluorescence lifetime data, presented here, can bridge studies in basic cell biology and systems biology. In systems biology studies, by monitoring fluorescent markers tethered to nascent RNA sequences, it was discovered that gene expression levels in single cells are not steady and uniform, but occur in periodic bursts[47]. It was further demonstrated that the bursts of RNA synthesis play an important role in coordination of gene networks[48] and contribute to regulation of basic cellular processes including differentiation, survival and maintenance of cellular homeostasis[49–51].

The FLIM approach described in this study can contribute to this field by quantitative detection of pulses of RNA synthesis. We propose that an increase in the concentration of proteins in nucleoli reflects burst in rRNA production in this nuclear organelle. This view is supported by previous studies of our and other groups. Recently, we reported on the bursts of rRNA synthesis in nucleoli by using label-free Raman spectroscopy—another optical approach that analyzes inelastic light scattering from cellular biomolecules, such as proteins and RNA. We found that bursts in rRNA synthesis timely coincide with massive recruitment of proteins to this nuclear organelle, producing protein concentration changes in a range of ~20–40 mg/mL, within several minute intervals[52,53]. These massive redistributions were confirmed by monitoring the signal from fluorescence-tethered proteins, which drive synthesis and processing of rRNA in nucleoli[54]. In this study, we utilized BSA for calibration of fluorescence lifetime response to the proteins concentration changes (Fig. 1). It is known that refractive index increments, which are constants that specify the variation of the RI with the solute concentrations, are close for most proteins[55]. For instance, RI increment for BSA is reported to be between 0.183 to 0.19, while averaged RI increment for cellular proteins is in the 0.185–0.19 range[55,56]. This indicates that the changes in BSA concentration and changes in concentration of intracellular proteins have a similar effect on the fluorescence lifetimes of EGFP and tdTomato.

An important and surprising finding of our study is the fluctuations of protein concentrations in the nuclear speckles. To our knowledge, this is a first study of global changes of protein condensation in this nuclear organelle. As discussed above, nuclear speckles do not contain any significant number of active genes and serve as a storage organelle. We suggest that the cyclic increases in protein concentration in the nuclear speckles manifest a withdrawal of RNA Polymerase II machinery from the nucleoplasm. This withdrawal may coincide with a short-term reduction of pre-mRNA output occurring between bursts of its synthesis. At the same time, cyclic bursts in pre-mRNA synthesis involve recruitment of the proteins from the nuclear speckles to active genes, which would result in the reduction of protein concentration in these nuclear organelles.

A major discovery in our work is that concentrations of proteins in the nuclear speckles and in the nucleoli are frequently changed in unison. To explain this synchrony in the cycles of proteins condensation into nuclear organelles and their release back to nucleoplasm, we propose the following mechanism. At any given moment, the concentration of the proteins in the nuclear organelles is defined by an equilibrium between the net condensation and release of proteins from the nuclear organelles, as discussed above (Fig. 5a). This equilibrium could change with onset of rRNA synthesis burst in nucleolus. At this stage, many

proteins involved in the synthesis and processing of nucleolar RNA and ribosomal assembly are redistributed from the nucleoplasm to the nucleolus[52,53]. As these proteins are recruited to nucleolus, their volume is replaced with water, which leads to a corresponding increase in the dielectric constant $\varepsilon$ of the nucleoplasm. These changes sequentially increase a difference in polarity between nuclear organelles and nucleoplasm, thus raising an energetic benefit of protein condensation into the organelles.

As a result, the equilibrium between the coalescence and diffusion-out of the proteins changes, leading to an accumulation of the proteins in the nuclear speckles, as demonstrated in the scheme (Fig. 5b). Finally, when the burst of RNA synthesis in the nucleolus ends, the subset of hydrophobic proteins, participating in RNA synthesis/processing, is released from the nucleolus. This redistribution reduces the dielectric constant of the nucleoplasm, which decreases the energetic benefits for condensation of proteins in the nuclear speckles, and other nuclear organelles leading to their release in the nucleoplasm (Fig. 5c). Changes in the protein concentrations in the nucleoli and nuclear speckles are accompanied with the corresponding changes of fluorescence lifetime of fluorophore probes in these nuclear organelles (Fig. 5d).

We propose that this synchronization mechanism in condensation and release of proteins in these nuclear organelles, serves for large-scale self-regulation of gene expression. An increase in the protein concentration in the nucleolus corresponds to bursts of rRNA production in this nuclear organelle[52], whereas an increase of proteins in the nuclear speckles apparently

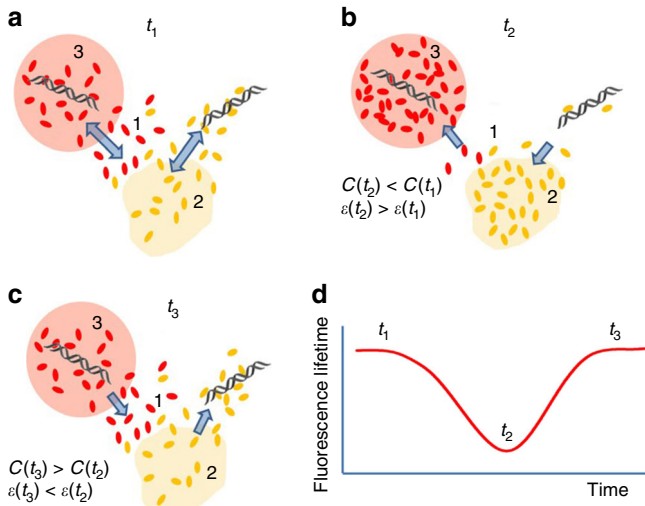

**Fig. 5** Model for synchronous proteins condensation in nucleoli and nuclear speckles. Schematic representation of the protein shuttling between the nucleoplasm (1), nuclear speckles (2), and nucleolus (3). **a** Time point $t_1$: Dynamic balance between the concentration of proteins in the nucleoplasm and nuclear organelles is maintained by condensation of proteins and their re-entrance of nucleoplasm. Nucleolar proteins are shown in red and proteins of the nuclear speckles are shown in yellow; equilibrium between proteins condensation and release is shown by arrows. **b** Time point $t_2$: Nucleolar proteins are recruited from nucleoplasm to nucleolus at the onset of rRNA synthesis burst. Reduced concentration of proteins (**C**), is accompanied by an increase in dielectric constant ($\varepsilon$) of nucleoplasm, which stimulates protein condensation in the nuclear speckles. **c** Time point $t_3$: at the end of rRNA synthesis burst, nucleolar proteins exit the nucleolus. Increased proteins concentration in the nucleoplasm is accompanied with decrease of $\varepsilon$, which triggers release of the proteins from the nuclear speckles. **d** Correspondence between the fluorescent lifetimes of fluorescent proteins in the nuclear organelles and the $t_1$–$t_3$ time points

indicates reduction of pre-mRNA synthesis. In such a mechanism, synchronous pulsations of protein concentrations in the nucleolus and the nuclear speckles manifest a time separation between the bursts of synthesis of rRNA in the nucleoli, and the bursts of synthesis of mRNA in the nucleoplasm, outside of these nuclear organelles. We hypothesize that this mechanism reduces the competition between the two RNA polymerase systems for cellular resources, leading to a higher efficiency of cellular metabolism.

In conclusion, we introduce a facile and non-invasive FLIM approach for quantitative monitoring of proteins condensation that enables discoveries in cellular protein reorganization processes. To our knowledge, FLIM approach is unparalleled in its capability to monitor accumulation and release of proteins in subcellular compartments. We report that the balance between the condensation and the release of proteins in phase-separated nuclear organelles is not stable, but drastically fluctuates in time. This dynamic process produces changes in the proteins concentration up to 100 mg/mL in single droplet organelles within several tens of minutes. We propose that synchronous changes in protein concentrations measured in the nucleoli and the nuclear speckles manifest a mechanism for temporal separation of the global bursts in rRNA and mRNA synthesis.

## Methods

**Cell culture and transfection**. CCL-2 HeLa cells (ATCC, Manassas, USA) were grown in glass-bottom dishes (Mattek, Ashland, USA) and cultured in Advanced DMEM (ThermoFisher Scientific, Grand Island, USA), supplemented with 2.5% fetal calf serum (Sigma-Aldrich, St. Louis, USA) at 37 °C in a humidified atmosphere containing 5% $CO_2$. The transfection of Fibrillarin-tdTomato, ASF/SF2 –EGFP, Coilin-EGFP, H2B-EGFP (Addgene, Cambridge, USA) was performed by using Lipofectamine LTX with Plus Reagent (Invitrogen, Carlsbad, USA) according to manufacturer's instructions and fluorescence lifetime measurements were performed within 12–24 h after the start of transfection procedure. For the FLIM imaging, morphologically representative cells exhibiting moderate fluorescence signal were selected. Each experimental group contained at least 20 cells. During imaging, cells were maintained under physiological conditions using a microscope stage incubator (Okolab, Burlingame, USA) and an objective heater (Okolab, Burlingame, USA).

For calibration of FLIM set-up cells with fluorescent proteins were fixed in 4% PFA/PBS for 15 min, permeabilized in 0.5% Triton X-100 and immersed into BSA solutions. This approach immobilizes fluorescent proteins in cells and removes membranes to allow access of BSA to the cell interior. In the outcome, the concentration of the BSA protein in the cell nucleus is adjusted for calibration.

In control experiments, the fluorescence lifetimes of histone H2B-EGFP, which binds to DNA and does not accumulate in the nuclear organelles, were also monitored. Additionally, cells expressing ASF/SF2- tdTomato were fixed with 2% formaldehyde in PBS, placed at 37 °C and monitored by FLIM, under the same conditions as the live cells for control measurements.

**FLIM approach**. FLIM monitoring of protein concentration is based on an inverse quadratic relation between the fluorescence lifetime of a fluorophore and the local RI[57]. Specifically, the fluorescence lifetime ($\tau$), is determined as a function of radiative and nonradiative pathways of fluorophore transition from excited to the ground level, as described by the equation: $1/\tau = k_r + k_{nr}$; where $k_r$ and $k_{nr}$ are the radiative and nonradiative decay rates, respectively. The $k_r$ is known as a function of the RI. The relation between fluorescence lifetime, ($\tau$), and the RI in the fluorophore's vicinity, ($n$), is described by the Strickler-Berg equation.

$$1/\tau = 2.880 \times 10^{-9}\ n^2\ \frac{\int I(\tilde{\nu})d\tilde{\nu}}{\int \tilde{\nu}d\tilde{\nu}} \int \frac{\varepsilon(\tilde{\nu})}{\tilde{\nu}}d\nu,$$ ($I$) is the fluorescence emission intensity,

($\varepsilon$) is the extinction coefficient, and ($\tilde{\nu}$) is the wavenumber[58].

In all experiments, the fluorescence lifetime images were recorded using DCS-120 confocal scanning FLIM system based on a TCSPC module (SPC-150, Becker & Hickl, Berlin, Germany). The scanner was attached to an inverted microscope (ECLIPSE TE2000-E, Nikon, Japan). Single photon fluorescence from the sample was collected through a Plan APO 100 × /NA1.4 oil immersion objective. The signals were detected by Hybrid Photon Detectors HPM-100-40 (Becker & Hickl, Berlin, Germany) that connected to a TCSPC module (Becker & Hickl, Berlin, Germany). The excitation source was a picosecond super continuum (400–650 nm) laser with an acousto-optic tunable filter (AOTF) (SC400-4, Fianium, UK) with acousto-optic tunable filter systems (Fianium Ltd., UK) and repetition frequencies of 80 MHz. The fluorescence signals were collected using band-pass filters HQ495LP for EGFP, and HQ620/60 for tdTomato (Chroma Technology Corp., Bellows Falls, USA). The minimum time channel width of the TCSPC module was

813 fs and the response time of the whole system was less than 30 ps. All images were acquired at 256 × 256 pixels and the typical integration time for a single image was 10–15 s. Lifetime calculations and fitting were performed using SPCImage software (Becker & Hickl, Berlin, Germany). To calculate the fluorescence lifetimes in the nuclear speckles, nucleoli, HP1 β heterochromatin domains and the Cajal bodies, studied nuclear organelles were manually segmented in the SPCImage software. Unless otherwise specified, the segmentation was applied to select all organelles of the same type in the cell at each type point. To calculate fluorescence decay of H2B-EGFP, the entire cell nucleus was manually selected. The lifetimes were calculated with 2 × 2 binning and FLIM images were automatically generated by SPCImage software. To facilitate image analysis, the brightness levels were adjusted to show fluorescence lifetimes in the sites with the highest photon count, corresponding to the labeled nuclear organelles.

Generation of plots, calculation of standard deviations and Pearson correlation coefficients were performed using Origin software (OriginLab, Northampton, USA).

In the control experiments the RI was measured in PBS, as well as in BSA/PBS solutions (50, 100, 150, and 200 mg/mL), using a conventional V-block refractometer and a 538 nm light source. Next, the fluorescence lifetimes of GFP in these solutions were measured (Supplementary Fig. 9). These experiments confirmed an inverse quadratic relation between the RI of the solution and $\tau$, in accordance with the Strickler-Berg equation.

**Reporting summary**. Further information on experimental design is available in the Nature Research Reporting Summary linked to this article.

## Data availability

The data that support the findings of this study are available from the corresponding authors upon reasonable request.

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

## Acknowledgements

S.M.L., P.X., T.Y.O., J.Q. were supported by the National Basic Research Program of China (2015CB352005); the National Natural Science Foundation of China (61525503/61620106016/61705142/61875135/61835009/81727804); Guangdong Natural Science Foundation Innovation Team (2014A030312008); Natural Science foundation of Guangdong Province (2017A030310136); Hong Kong, Macao, and Taiwan cooperation innovation platform & major projects of international cooperation in Colleges and Universities in Guangdong Province (2015KGJHZ002); and Shenzhen Basic Research Project (JCYJ20150930104948169/JCYJ20160328144746940/JCYJ20170412105003520/JCYJ20170818090620324).

## Author contributions

A.P., S.M.L., L.L., J.Q. and P.N.P. conceived the study and designed experiments. A.P., S.M.L., X.P. and A.N.K. performed experiments. A.P., S.M.L., T.Y.O., I.R., J.Q. and P.N.P. analyzed data. A.P., S.M.L., T.Y.O., J.Q. and P.N.P., wrote the manuscript. All authors read and accepted the manuscript.

## Additional information

**Competing interests:** The authors declare no competing interests.

