## [Peer Review File · Nature Communications]

Reviewers' comments:

Reviewer #1, an expert in FLIM (Remarks to the Author):

This is an intriguing and elegant paper where the authors observe periodic fluctuations in the fluorescence lifetime of fluorophores labelling proteins expected to be localised in nucleoli. They propose that the variations in fluorescence lifetime are due to fluctuations in the local refractive index and suggest that this may be due to proteins transferring in and out of nucleoli.

To play devil's advocate, I note that the fluorescence lifetime can be a sensitive function of many local environmental factors, such as temperature and pH, and that a global variation on one of these factors might also explain the observed oscillations in fluorophore lifetime – particularly since they seem to be synchronised.

Figure S1 shows a small periodic variation in the lifetime of H2B-EGFP. This serves as a negative control since the observed variations in lifetime of fluorophores in the nuclear organelles is larger. However, the presence of the periodic observation in the H2B-EGFP lifetime suggests that there may be global changes in refractive index in the nucleus or other environmental factors varying over time to modulate the EGFP index. Was the temperature monitored over the experiments and did it show any periodic variation? Is it possible that changes in temperature might result in changes in refractive index? Were the fluorescence lifetime measurements made through a "magic angle" polariser to eliminate potential anisotropy artefacts? Temperature changes can modify the fluorescence polarisation anisotropy of GFP (Donner et al, NanoLett 2012; [dx.doi.org/10.1021/nl300389y](https://doi.org/10.1021/nl300389y)). Fluorescence anisotropy could also be impacted by molecular crowding.

The calibration experiments with BSA indicate that the lifetimes of the GFP and tdTomato-labelled constructs vary with BSA concentration. The gradient of change in lifetime with BSA concentration is the same for all the constructs and supports the explanation of refractive index changes. It would have been helpful if the refractive index of these BSA solutions could be measured to confirm the observed lifetime changes agree with the Strickler-Berg equation.

If the change in lifetime is due to a change in refractive index, it does not seem safe to assume that the BSA data can be used to provide a calibration to infer the absolute concentrations of other proteins from lifetime measurements. As well as proving that the variation in lifetime is due to the refractive index changes, it would be necessary to establish the relationship between refractive index and protein concentration. A set of measurements of refractive index as a function of concentration of solutions of these protein would be useful. However, extrapolation to the nuclear organelles may be complicated by the macromolecular crowding (i.e. excluded volume) phenomenon.

The change in lifetime of the fluorophores in the nuclear organelles may be changing because local or global variations of environmental factors other than refractive index, for example, pH or temperature. It would be helpful to have a fluorophore present in the nucleoplasm reporting any background variations in the same cells with the reporter in the nuclear organelles.

For figure 2, I am confused as to whether the EGFP signal is limited to the nuclear speckles or is also present in the nucleoplasm as indicated in the text. If this is the case, why is GFP not visible across the nucleus? How were the nuclear speckles segmented?

In figure 3, why are there somewhat punctate features of shorter lifetime? Did the segmentation average lifetime over each whole nucleolus? What are the compartments in the nucleoli that appear in the intensity image of figure 3A? It might be interesting to segment over these. It could also be useful to show the evolution of the fluorescence lifetime histograms corresponding to each nucleoli. Here the punctate variations in lifetime suggest that averaging over the whole organelle

to calculate a mean lifetime shift and then attribute that to a mean change in RI and therefore protein concentration may not be appropriate.

Reviewer #2, an expert in liquid-liquid protein phase separations (Remarks to the Author):

I have reviewed the manuscript NCOMMS-18-14493 titled "Reorganization of nuclear organelles through the cycles of protein condensation and discharge by fluorescence lifetime imaging" by Pliss et al.

The authors have used fluorescence lifetime imaging (FLIM) to monitor real-time protein concentration changes of condensed nuclear organelles in live cultured cells. The concentration monitoring relies on the lifetime dependence of the fluorescent probes to the square of the refractive index of the environment, which in turn is a function of protein concentration. Their main conclusions are the following: 1. Changes in protein concentration were observed in major nuclear organelles: nucleoli, nuclear speckles, Cajal bodies, and clusters of heterochromatin. 2. Protein concentration changes were synchronous for different organelles of the same cell. 3. A molecular mechanism is proposed for coordinating synchronous accumulation of proteins in nuclear organelles and may apply to cellular metabolism and coordination of gene expression.

In principle, this manuscript could be appropriate for Nature Communications, but only after additional experiments and major revisions to address a number of significant issues. The reason for this recommendation is that the article indicates very interesting and potentially significant results, particularly the oscillatory dynamics. But much of the data are not wholly convincing, and possible artifacts have not been ruled out. Additional supporting controls and experiments are necessary. Moreover, the overall presentation could improve.

Specific comments:

Strengths

1. The fluorescent proteins show a clear linear dependence of the fluorescent lifetime on the concentration of the (calibration) protein BSA. This linear dependence is strong and consistent for all probes tested.
2. The fluorescence lifetimes of the probe proteins very clearly vary as a function of time and in fact show potentially very interesting oscillatory behavior, suggesting oscillatory possible protein concentration levels in the organelles

Weak Points/Questions (that need to be addressed in the revised version):

1. One of the most important controls is the histone H2B-EGFP that is said to bind to DNA and not accumulate in the nuclear organelles. It would be helpful if this were included on the main plots, especially so we can assess the relative changes compared to the oscillations seen in the other organelle-accumulating fluorescent proteins. While I agree that the control appears relatively "static", it actually does show some oscillations, which could indicate that the FLIM System (Laser/detectors etc.) exhibit some intrinsic oscillations. Indeed, the timescale of the oscillation in figure S1 is on the order of 10min, which is *very* similar to the oscillation period that the others claim they are detecting in living cells. Quantitative metrics for the intrinsic noise/oscillations in the FLIM system are sorely needed to dispel worries that these data reflect an artifact.
2. While the linear relationship between the fluorescent lifetime and the concentration of control BSA is convincing, and in combination with the Strickler-Berg relationship we can assume this is due to the dependence of fluorescent lifetime on refractive index, it would be more convincing to characterize both the refractive index of the BSA solutions and the response of the fluorescent

probes in other solutions of known refractive index. Especially since this is a relatively new way of monitoring concentration changes in cells. There are many other effects that the environment can have on the photophysics of the probes, including dependence on viscosity and electrostatics. This can become especially complicated if the fluorophore is bound in a protein, although I am aware that the dependence of lifetime on refractive index has been directly verified for GFP. This relationship holds true only when k_{nr} remains constant.

3. A main worry is the significance of the resulting lifetime changes compared to the size of the errors on the calibration plot. Although they are difficult to see, it looks like one standard deviation can be about 40 ps, which is about the same size as the fluctuations of the lifetimes in the cells. Again, the paper needs additional quantitative metrics that indicate the values and ranges of the calibration compared to those of some of the results and the control and some assessment of the significance.

4. The description of the calibration of the protein fusions is confusing. Clarification is needed on what we are expecting to measure when the cell is fixed and permeabilized? Are you measuring the extracts of the fusion proteins when they leave the cell and enter the BSA solutions? It would be helpful to explicitly state for non-experts in cell handling, what the procedures give us rather than just what the procedure was.

5. The manuscript overall could benefit from many additional references as there are many statements that are taken for granted or the word and/or referencing is misleading.

a. For example, page 5 paragraph 3: "nuclear speckles were visualized by the signal from ASF/SF2-EGFP". References and possibly additional diagrams for these procedures/strategies will be helpful for a wider audience reading Nature Communications.

b. Page 2, paragraph 2: "a key factor driving the formation and maintenance of nuclear organelles ... water is highly polar as compared to the surface of the molecules ..." The reference cited in JCTC predicts the ranges of dielectric "constants" in the protein and surface, but the driving force of organelle formation is not discussed in that reference and may not be so simple.

c. "Until now, studies investigating kinetics of proteins condensation into droplet organelles were mostly performed in solutions containing several isolated nuclear proteins, as well as inert macromolecules added for simulation of the intercellular molecular crowding" The crowding review cited does not mention its relation to organelle condensation at all, other more relevant citations should be included.

d. Page 11 paragraph 3 regarding "theoretical considerations"—can you substantiate these strong statements?

e. Page 6: "a quantitative model...has to address to two opposite processes: the condensation of proteins into droplets and their release back to the nucleoplasm..." This statement does not really make a lot of sense, based a large amount of literature on the biophysics of intracellular phase separation.

f. Page 2: "...water is highly polar compared to the surface of protein macromolecules: this difference leads to proteins condensation and separation from the water based nucleoplasm". This statement is overly simplistic and largely incorrect. Intracellular condensates are open structures full of water.

6. Page 5 Paragraph 2: "Studied organelles were segmented, and the fluorescence lifetime values for the same type of organelles were averaged for each time point". It is also a little confusing regarding what exactly is being averaged...how are you defining a segment? Is it the entire organelle? A diagram may be helpful here.

Grammatical and presentation suggestions:

1. Ref #22 and #23 are the same.

2. Page 4 Paragraph 3: proteins "located into" the nuclear organelles. These experiments were not yet in the nuclear organelles but this wording seemed to suggest they would be. Please clarify.

3. I suggest significant improvement to the quality of the figures, specifically resolution, sizes of boxes (should be consistent) and alignment of labels (for a professional publication, at least the labeling of panels should lie in the same line). The error bars (which I presume are present on Figure 1) are not really visible and the figures appear to be low resolution.

4. PBS is abbreviated but not written in full the first time.
5. Reference formatting is inconsistent.
6. Page 8, line 6: "involved into" ?
7. There are numerous dashes that seem unnecessary. Page 4 paragraph 3: "label - nucleoli, nuclear speckles..." and page 8: "coilin-EGFP, - a marker protein". Page 11 paragraph 2: "water, - around 80".

Reviewer #1, an expert in FLIM (Remarks to the Author):

Q1.1. This is an intriguing and elegant paper where the authors observe periodic fluctuations in the fluorescence lifetime of fluorophores labelling proteins expected to be localised in nucleoli. They propose that the variations in fluorescence lifetime are due to fluctuations in the local refractive index and suggest that this may be due to proteins transferring in and out of nucleoli.

To play devil's advocate, I note that the fluorescence lifetime can be a sensitive function of many local environmental factors, such as temperature and pH, and that a global variation on one of these factors might also explain the observed oscillations in fluorophore lifetime – particularly since they seem to be synchronised.

A1.1. It is true that the fluorescence lifetime can be sensitive to many local environmental factors and the impact of local environment on fluorescence lifetime of fluorophores should be considered. However, it is commonly accepted that lifetime of fluorescent proteins is less sensitive and more stable in different environments, because the fluorophore is located inside rigid β -barrel domain, which significantly shields it from the surrounding media, dampening its influence. In our experiments, we ruled out any significant influence of pH and temperature on fluorescence lifetime by our experimental conditions, as discussed below (see answers A1.3 and A1.7). In the revision, we emphasize the significance of RI as a parameter, which is unambiguously connected with the fluorescence lifetimes of fluorescent proteins (Page 3, last paragraph).

Q1.2. Figure S1 shows a small periodic variation in the lifetime of H2B-EGFP. This serves as a negative control since the observed variations in lifetime of fluorophores in the nuclear organelles is larger. However, the presence of the periodic observation in the H2B-EGFP lifetime suggests that there may be global changes in refractive index in the nucleus or other environmental factors varying over time to modulate the EGFP index.

A1.2. We agree with the Reviewer that the variations of H2B- EGFP suggest global variations of the refractive index in the cell nucleus. For instance, an exchange of soluble proteins between the nuclear organelles and nucleoplasm, cellular growth, balance between proteins synthesis and proteins decay,- all these factors can influence the concentration of the soluble proteins in the nucleoplasm and cause change in the lifetime of fluorescent proteins in the cell nucleus. In addition, the histone H2B-EGFP lifetime could be influenced by local changes in DNA compaction.

Thus, small fluctuations in the lifetime of H2B-EGFP could be expected. At the same time, the observed changes (~5 - 15 ps per 60 min observation) were significantly smaller than the corresponding changes in the nuclear organelles. This difference in fluctuations highlights significance of our control experiments.

In addition, an instrument measurement error could influence our data. We addressed this issue and established an instrument measurement error. In these experiments, we measured the lifetime of fluorescent proteins in formaldehyde fixed cells and found similar variations ($\Delta\tau \leq 7\text{ps}$) (Fig. S1).

In revision, an additional discussion on H2B-EGFP measurements is provided in the text (Page 6, second paragraph).

Q1.3. Was the temperature monitored over the experiments and did it show any periodic variation? Is it possible that changes in temperature might result in changes in refractive index?

A1.3. During imaging, cells were kept in the Okolab incubator, the objective heater was also applied. Under these conditions, the temperature variations do not exceed 0.1 degrees, according to the manufacturer specifications. In revision, we provided a description of temperature control (page 17, paragraph 1). We also performed additional control experiment, wherein lifetimes of fluorescence proteins were monitored in formaldehyde-fixed cells placed at 37C. We observed no significant fluorescence lifetime changes in these cells (Fig. S1).

Q1.4. Were the fluorescence lifetime measurements made through a “magic angle” polariser to eliminate potential anisotropy artefacts? Temperature changes can modify the fluorescence polarisation anisotropy of GFP (Donner et al, NanoLett 2012; [dx.doi.org/10.1021/nl300389y](https://doi.org/10.1021/nl300389y)).

Fluorescence anisotropy could also be impacted by molecular crowding.

A1.4. We appreciate the idea that the fluorescence polarization anisotropy can be a productive approach for probing the cellular environment (similarly to the FLIM approach proposed in our study). We are aware that anisotropy artifacts can potentially affect lifetime measurements but believe that in our experiments the influence of the potential anisotropy artifacts on FLIM is negligible. In general, to be sensitive for anisotropy, experimental imaging geometry requires a polarizer, or polarizing beamsplitter, in the detection arm for single or two-detector (simultaneous) polarization-resolved measurements, respectively. It is also assumed that the excitation laser light is linearly polarized. According to the Perrin's equation, fluorescence anisotropy r is defined as $r = r_0 / (1 + \tau_{fl} / \phi)$, where r_0 is the initial anisotropy, a fixed molecular property given by the orientation of the absorption and emission transition dipole moments, ϕ is rotational correlation time and τ_{fl} is fluorescence lifetime. Consequently, τ_{fl} can be presented as a function of anisotropy r . As it was shown both theoretically and experimentally by Fixler and co-workers (Fixler et al, 2006), an incorrect arrangement of τ_{fl} measurement without consideration of its dependence on anisotropy may yield an artificial decay contribution and the overall measured signal might be wrongly interpreted as multiexponentially decaying emission. An artifact decay time can be determined as $\tau_a = \tau_{fl} r / r_0$. To avoid appearance of the artificial decay contribution, the use of a polarizer in the emission detection channel is proposed, which is at “magic angle” (e.g., 54.7°) relatively to the linearly polarized excitation light (Fixler et al, 2006). However, in our FLIM experiments that were performed in the absence of polarizers/analysers for excitation and emission, we did not observe any additional decay, fluorescence decays of the FPs were strictly monoexponential, supporting absence/negligibility of the anisotropy artifacts.

It should be noted, that in the case of microscopy the depolarization occurs in the optical train of the system, particularly when using high numerical aperture objectives, which significantly reduce the r_0 value (Levitt et al, 2009). Moreover, the use of high aperture objectives, which are not intended for the polarized light applications, may result in depolarization of the excitation and emission (Erdelyi M et al, 2014) and, correspondingly, diminishing any possible effect of the anisotropy artifacts.

References

1. Levitt JA et al., (2009) Fluorescence lifetime and polarization-resolved imaging in cell biology Current Opinion in Biotechnology, 20:28–36
2. Fixler D et al (2006) IEEE Trans Biomed Eng. 53:1141-1152
3. Miklos Erdelyi M et al (2014) Analyzing Receptor Assemblies in the Cell Membrane Using Fluorescence Anisotropy Imaging with TIRF Microscopy, Plos One, 9: e100526

Q1.5. The calibration experiments with BSA indicate that the lifetimes of the GFP and tdTomato-labelled constructs vary with BSA concentration. The gradient of change in lifetime with BSA concentration is the same for all the constructs and supports the explanation of refractive index changes. It would have been helpful if the refractive index of these BSA solutions could be measured to confirm the observed lifetime changes agree with the Strickler-Berg equation.

A1.5. To address this concern, we performed the following experiments. First, we measured the refractive indices for different concentrations of BSA/PBS solutions. Then we measured the fluorescence lifetime of GFP in these solutions. The obtained data confirm an inverse quadratic relation between the fluorescence lifetime of GFP and RI of solutions, predicted by Strickler-Berg equation. These measurements were included in the Supplementary material (Fig. S9).

Q1.6. If the change in lifetime is due to a change in refractive index, it does not seem safe to assume that the BSA data can be used to provide a calibration to infer the absolute concentrations of other proteins from lifetime measurements. As well as proving that the variation in lifetime is due to the refractive index changes, it would be necessary to establish the relationship between refractive index and protein concentration. A set of measurements of refractive index as a function of concentration of solutions of these protein would be useful. However, extrapolation to the nuclear organelles may be complicated by the macromolecular crowding (i.e. excluded volume) phenomenon.

A1.6. BSA was used for calibration, because the relation between the BSA solutions concentration and solution refractive index is close to that for averaged cellular proteins.

In general, the concentration of solute and the refractive index of the solution are linked through the refractive index increment in accordance to the equation: $n_o = n_w + \alpha C$. In this equation, n_o is the RI of the solution, n_w is the RI of the solvent, C is the concentration of solute (proteins) (g/cm^3), and α is the refractive index increment of the solute, which determines the change of n_o with change of C .

Averaged refractive index increments for cellular proteins were reported in the range from 0.1845 to 0.19 in green and yellow light (Zhao et al., 2011). Refractive increments for a variety of specific proteins fall within $\sim 2\%$ of the mean value (Barer et al., 1954).

Consistent with that, refractive index increment of BSA varies from 0.183 to 0.19 depending on the measurement method and perhaps the purity of the solute (Tumolo et al., 2004). Considering that the refractive increment of BSA is very close to that of averaged cellular proteins, we believe that BSA is a suitable protein for concentration calibration experiments shown in Fig 1 a-b.

In the Revision, we clarified the suitability of BSA for concentration calibration (page 13, paragraph 2).

References

- 1) Zhao HY et al. On the Distribution of Protein Refractive Index Increments. *Biophys J.* 2011; 100: 2309-17
- 2) Barer, R., and Joseph, S. Refractometry of Living Cells. *Journal of Cell Science.* 1954 s3-95: 399-423
- 3) Tumolo T., et al., Determination of the refractive index increment (dn/dc) of molecule and macromolecule solutions by surface plasmon resonance. *Analytical Biochemistry.* 2004; 333: 273-279.

Q1.7. The change in lifetime of the fluorophores in the nuclear organelles may be changing because local or global variations of environmental factors other than refractive index, for example, pH or temperature. It would be helpful to have a fluorophore present in the nucleoplasm reporting any background variations in the same cells with the reporter in the nuclear organelles.

A1.7. Up to date, RI is determined as the only factor in a cellular environment, changes in which clearly correlate with changes in the fluorescence lifetime of fluorescent proteins in the absence of energy transfer. This dependence has been validated previously by theoretical and experimental studies (Sugling et al, 2002; Toptygin 2003; Borst et al, 2005; van Manen et al, 2008).

With regard to pH, it does not influence noticeably the fluorescence lifetime of fluorescent proteins. The fluorescence lifetime of GFP has been shown to be stable across the broad pH range under 480 ± 20 nm excitation (Kneen et al 1998). A more detailed discussion of the fluorescent proteins lifetime stability at different pH is given in our earlier study (Pliss et al, 2012).

The temperature, as discussed above, was strictly controlled by the incubator equipped with an objective heater - (page 17 paragraph 1). To further rule out any potential impact of temperature fluctuations, we made an additional control experiment. Cells expressing ASF/SF2-EGFP were fixed in formaldehyde, placed in the stage-mounted incubator, at 37C, and monitored by FLIM. As could be expected, we did not find any significant changes in fluorescence lifetime in the fixed cells (Fig. S1).

To address this concern, we highlighted the link between the fluorescence lifetime of fluorescent proteins and the RI of their environment (Page 3, last paragraph).

References

- 1) Kneen, M et al., (1998) Green fluorescent protein as a noninvasive intracellular pH indicator, *Biophysical Journal* 74, 1591-1599.
- 2) Pliss A, et al., (2012) Fluorescence Lifetime of Fluorescent Proteins as an Intracellular Environment Probe Sensing the Cell Cycle Progression. *ACS Chemical Biology*, 8:1385-92.
- 3) Suhling K, et al., (2002) Imaging the environment of green fluorescent protein. *Biophys J.* 83:3589-95
- 4) Toptygin D. Effects of the solvent refractive index and its dispersion on the radiative decay rate and extinction coefficient of a fluorescent solute. *J Fluoresc.* 2003; 13: 201-19.
- 5) Borst J W et al. (2005) Effects of Refractive Index and Viscosity on Fluorescence and Anisotropy Decays of Enhanced Cyan and Yellow Fluorescent Proteins. *J Fluorescence*, 15:153-160
- 6) van Manen HJ et al. (2008) Refractive Index Sensing of Green Fluorescent Proteins in Living Cells Using Fluorescence Lifetime Imaging Microscopy. *Biophysical Journal* 94: L67-L69

Q1.8. For figure 2, I am confused as to whether the EGFP signal is limited to the nuclear speckles or is also present in the nucleoplasm as indicated in the text. If this is the case, why is GFP not visible across the nucleus? How were the nuclear speckles segmented?

A1.8. EGFP –ASF/SF2 is significantly accumulated in the nuclear speckles and, at lower concentrations, it is also present in the nucleoplasm. Typical distribution of this protein is shown on the fluorescence intensity image (Fig 2a, upper left panel).

FLIM images on the Figure 2 were prepared by intensity segmentation. In this approach, the intensity threshold was set to pick up only the brightest signal, which corresponds to nuclear speckles. At the same time, lifetime data in the surrounding nucleoplasm were not shown.

We included a detailed description of this technique in the revision of our manuscript (Page 18, paragraph 1).

Q1.9. In figure 3, why are there somewhat punctate features of shorter lifetime? Did the segmentation average lifetime over each whole nucleolus? What are the compartments in the nucleoli that appear in the intensity image of figure 3A? It might be interesting to segment over these. It could also be useful to show the evolution of the fluorescence lifetime histograms corresponding to each nucleoli. Here the punctate variations in lifetime suggest that averaging over the whole organelle to calculate a mean lifetime shift and then attribute that to a mean change in RI and therefore protein concentration may not be appropriate.

A1.9.

1) The FLIM data in Figure 3 suggests irregularities in the protein density in the nucleoli. It is known, that condensation of nucleolar proteins creates multiple nucleolar sub-compartments including the fibrillar centers (FC), dense fibrillar components (DFC), and granular components (Feric et al., 2016). Fibrillarin is mostly accumulated in the DFC, and to a lesser extent in FC. It is likely that the variations in proteins concentrations in different sub-compartments in nucleoli produce corresponding differences in local RI, which results into irregular (dotted) lifetime pattern.

2) The segmentation was applied to the entire nucleolus. In cells, which contained several nucleoli, all these organelles were segmented and the fluorescence lifetime values were averaged. It is important to note that lifetimes fluctuated synchronously in different nucleoli of the same cell. Thus, averaging helps to identify general trend in protein condensation for nucleoli (Fig. S5).

3) We attempted to monitor smaller clusters inside the nucleoli. Fluorescence lifetime changes in such clusters tend to coincide with the average changes in the nucleolus. However, in the small area histograms often have

irregular shape and therefore are difficult for analysis (Fig. S4). We believe that a specialized approach should be developed for the analysis of the sub-nucleolar fluctuations in fluorescence lifetimes. To address Q1.9 we discussed the FLIM images pattern (Page 8, last paragraph) and prepared additional supplementary material (Figure S4 and Figure S5).

Reviewer #2, an expert in liquid-liquid protein phase separations (Remarks to the Author):

I have reviewed the manuscript NCOMMS-18-14493 titled "Reorganization of nuclear organelles through the cycles of protein condensation and discharge by fluorescence lifetime imaging" by Pliss et al.

The authors have used fluorescence lifetime imaging (FLIM) to monitor real-time protein concentration changes of condensed nuclear organelles in live cultured cells. The concentration monitoring relies on the lifetime dependence of the fluorescent probes to the square of the refractive index of the environment, which in turn is a function of protein concentration. Their main conclusions are the following: 1. Changes in protein concentration were observed in major nuclear organelles: nucleoli, nuclear speckles, Cajal bodies, and clusters of heterochromatin. 2. Protein concentration changes were synchronous for different organelles of the same cell. 3. A molecular mechanism is proposed for coordinating synchronous accumulation of proteins in nuclear organelles and may apply to cellular metabolism and coordination of gene expression.

In principle, this manuscript could be appropriate for Nature Communications, but only after additional experiments and major revisions to address a number of significant issues. The reason for this recommendation is that the article indicates very interesting and potentially significant results, particularly the oscillatory dynamics. But much of the data are not wholly convincing, and possible artifacts have not been ruled out. Additional supporting controls and experiments are necessary. Moreover, the overall presentation could improve.

Specific comments:

Strengths

1. The fluorescent proteins show a clear linear dependence of the fluorescent lifetime on the concentration of the (calibration) protein BSA. This linear dependence is strong and consistent for all probes tested.
2. The fluorescence lifetimes of the probe proteins very clearly vary as a function of time and in fact show potentially very interesting oscillatory behavior, suggesting oscillatory possible protein concentration levels in the organelles

Weak Points/Questions (that need to be addressed in the revised version):

Q2.1.

1. One of the most important controls is the histone H2B-EGFP that is said to bind to DNA and not accumulate in the nuclear organelles. It would be helpful if this were included on the main plots, especially so we can assess the relative changes compared to the oscillations seen in the other organelle-accumulating fluorescent proteins. While I agree that the control appears relatively "static", it actually does show some oscillations, which could indicate that the FLIM System (Laser/detectors etc.) exhibit some intrinsic oscillations. Indeed, the timescale of the oscillation in figure S1 is on the order of 10min, which is *very* similar to the oscillation period that the others claim they are detecting in living cells. Quantitative metrics for the intrinsic noise/oscillations in the FLIM system are sorely needed to dispel worries that these data reflect an artifact.

A2.1.

1) Small changes in H2B-EGFP lifetime are expected in our experimental system. For instance, the RI of the cell nucleus is defined by a dynamic balance between the cellular volume, as well as the variations in the synthesis and decay of soluble proteins. Thus, any stochastic variations in the cellular growth or metabolism may influence the intracellular RI, and induce corresponding changes in the H2B-EGFP lifetime. The H2B-EGFP lifetime could

be also influenced by cyclic changes in the concentration of soluble proteins, as they are redistributed between nucleoplasm and nuclear organelles. At the same time, the fluorescence lifetime oscillations for H2B-EGFP (~5 - 15 ps per 60 min observation) were significantly smaller in comparison to fluorescence lifetime oscillations in the nuclear organelles (Fig. 2- Fig. 4).

We appreciate the suggestion to quantify the intrinsic noise of FLIM system. In response to this concern, we characterized an instrument measurement error. The fluorescence lifetime was sequentially measured in the formaldehyde fixed cells, kept at 37°C (Fig. S1). We found variations of 5 - 7 picoseconds per 60 min, which defines the instrument measurement error. We have shown the measurement error bar in Fig. 2b and Fig. 3b. In addition, we discussed oscillations of H2B-EGFP lifetime (Page 6, second paragraph).

Q2.2.

2. While the linear relationship between the fluorescent lifetime and the concentration of control BSA is convincing, and in combination with the Strickler-Berg relationship we can assume this is due to the dependence of fluorescent lifetime on refractive index, it would be more convincing to characterize both the refractive index of the BSA solutions and the response of the fluorescent probes in other solutions of known refractive index. Especially since this is a relatively new way of monitoring concentration changes in cells. There are many other effects that the environment can have on the photophysics of the probes, including dependence on viscosity and electrostatics. This can become especially complicated if the fluorophore is bound in a protein, although I am aware that the dependence of lifetime on refractive index has been directly verified for GFP. This relationship holds true only when k_{nr} remains constant.

A2.2. In response to this concern, we have plotted a correlation between BSA concentrations, refractive index and fluorescence lifetime of EGFP (Fig. S9).

We also discussed our work in the context of previous studies investigating the fluorescence lifetime of fluorescent proteins in a subcellular environment. Up-to-date, refractive index is the only factor, which is unambiguously coupled with the fluorescence lifetime of intracellular fluorescent proteins in the absence of FRET. Previously, we shown that fluorescence lifetime of GFP and tdTomato responds to changes of intracellular refractive index during cellular division, cellular growth, or cellular volume change in either hypotonic or hypertonic buffers (Pliss et al, 2012). In the revision of this manuscript, we refer these studies for clarification.

With regard to viscosity, previous studies in solutions have shown that the fluorescence lifetime of GFP correlates with the square of RI, and not with viscosity (Suhling et al 2002; Borst J W et al. 2005; van Manen HJ et al. 2008). In addition, in the cell nucleus, any changes in viscosity would be related to the changes in RI. Thus, calibration charts on the Figure 1 could be applied for FLIM data analysis.

With regard to electrostatic interactions, we believe that RI also covers this case. The electrostatic interactions of FPs with environment are interconnected with the dielectric constant of the medium, which directly determines the RI.

This issue has been additionally discussed and clarified (page 3, paragraphs 4 - 5)

References

- 1) Pliss A, et al., (2012) Fluorescence Lifetime of Fluorescent Proteins as an Intracellular Environment Probe Sensing the Cell Cycle Progression. ACS Chemical Biology, 8:1385-92.
- 2) Suhling K et al., (2002) The Influence of Solvent Viscosity on the Fluorescence Decay and Time-Resolved Anisotropy of Green Fluorescent Protein. J Fluorescence, 12:91-95
- 3) Borst J W et al. (2005) Effects of Refractive Index and Viscosity on Fluorescence and Anisotropy Decays of Enhanced Cyan and Yellow Fluorescent Proteins. J Fluorescence, 15:153-160
- 4) van Manen HJ et al. (2008) Refractive Index Sensing of Green Fluorescent Proteins in Living Cells Using Fluorescence Lifetime Imaging Microscopy. Biophysical Journal 94: L67-L69

Q2.3.

3. A main worry is the significance of the resulting lifetime changes compared to the size of the errors on the calibration plot. Although they are difficult to see, it looks like one standard deviation can be about 40 ps, which is about the same size as the fluctuations of the lifetimes in the cells. Again, the paper needs additional quantitative metrics that indicate the values and ranges of the calibration compared to those of some of the results and the control and some assessment of the significance.

A2.3. The standard deviations for measurements in solutions was ranging from 8 to 12 picoseconds. We corrected and enlarged Fig. 1. In addition, we performed control experiments by subsequent measurements of the fluorescent lifetime of ASF/SF2-EGFP in the formaldehyde-fixed cells. In the fixed cells, the difference between the minimum and the maximum fluorescence lifetimes was 5-7 picoseconds (Fig. S1).

Q2.4.

4. The description of the calibration of the protein fusions is confusing. Clarification is needed on what we are expecting to measure when the cell is fixed and permeabilized? Are you measuring the extracts of the fusion proteins when they leave the cell and enter the BSA solutions? It would be helpful to explicitly state for non-experts in cell handling, what the procedures give us rather than just what the procedure was.

A.2.4. We performed calibration of the fluorescence lifetimes of fluorescent proteins and BSA concentrations both in solutions and in the cells. Calibration in BSA/PBS solutions was made using commercially available tdTomato and EGFP. Cellular calibration was made by expressing either H2B-EGFP, Coilin-EGFP, HP1beta-EGFP, ASF/SF2-EGFP or Fibrillarin-tdTomato protein fusions in the cultured cells.

In our cellular calibration experiments, we made the following steps: (i) Fix fluorescent proteins inside the cells; (ii) Open membranes by permeabilization to allow diffusion of BSA into the cell; (iii) Immerse cells in solutions with different BSA concentrations and measure their lifetimes.

This approach enables BSA to enter the cell and change the RI in the environment of fluorescent protein. We clarified our methods in the revised manuscript (page 17 paragraph 2).

Q.2.5.

5. The manuscript overall could benefit from many additional references as there are many statements that are taken for granted or the word and/or referencing is misleading.

a. For example, page 5 paragraph 3: "nuclear speckles were visualized by the signal from ASF/SF2-EGFP". References and possibly additional diagrams for these procedures/strategies will be helpful for a wider audience reading Nature Communications.

A.2.5a. Additional description and references have been provided (Page 6, paragraph 3 and page 7, paragraph 1). Reference 36 contains an overview of fluorescence proteins applications in cell biology, and references 11 and 34 contain experimental studies of ASF/SF2, fibrillarin, H2B and HP1 fluorescent fusions in live cell nucleus.

b. Page 2, paragraph 2: "a key factor driving the formation and maintenance of nuclear organelles ... water is highly polar as compared to the surface of the molecules ..." The reference cited in JCTC predicts the ranges of dielectric "constants" in the protein and surface, but the driving force of organelle formation is not discussed in that reference and may not be so simple.

A.2.5b. We agree that the formation of organelles is a complex process that can be driven by multiple forces. In this context, the difference in dielectric constants/polarities of proteins and water can be one of the factors driving the formation of the nuclear organelles.

We clarified our statement, and corresponding changes were introduced into the text (Page 2, paragraph 2).

c. “Until now, studies investigating kinetics of proteins condensation into droplet organelles were mostly performed in solutions containing several isolated nuclear proteins, as well as inert macromolecules added for simulation of the intercellular molecular crowding” The crowding review cited does not mention its relation to organelle condensation at all, other more relevant citations should be included.

A.2.5c. We modified this sentence and included recent and more relevant references (Page 3, paragraph 3. Ref 9, 18, 19, 20).

d. Page 11 paragraph 3 regarding “theoretical considerations”—can you substantiate these strong statements?

A.2.5d. This statement was removed.

e. Page 6: “a quantitative model...has to address to two opposite processes: the condensation of proteins into droplets and their release back to the nucleoplasm...” This statement does not really make a lot of sense, based a large amount of literature on the biophysics of intracellular phase separation.

A.2.5e. This statement refers to dynamic shuttling of the soluble proteins between the nucleoplasm and the nuclear organelles (Phair and Misteli 2000). We have changed the paragraph to clarify this.

Reference

Phair RD, Misteli T. High mobility of proteins in the mammalian cell nucleus. *Nature* 404, 604-609 (2000).

f. : this difference leads to proteins condensation and separation from the water based nucleoplasm”. This statement is overly simplistic and largely incorrect. Intracellular condensates are open structures full of water.

A.2.5f. We have removed this sentence.

6. Page 5 Paragraph 2: “Studied organelles were segmented, and the fluorescence lifetime values for the same type of organelles were averaged for each time point”. It is also a little confusing regarding what exactly is being averaged...how are you defining a segment? Is it the entire organelle? A diagram may be helpful here.

A.2.6. Additional description was included (Page, 18, paragraph 1)

Grammatical and presentation suggestions:

1. Ref #22 and #23 are the same.

The error was corrected.

2. Page 4 Paragraph 3: proteins “located into” the nuclear organelles. These experiments were not yet in the nuclear organelles but this wording seemed to suggest they would be. Please clarify.

The sentence was clarified.

3. I suggest significant improvement to the quality of the figures, specifically resolution, sizes of boxes (should be consistent) and alignment of labels (for a professional publication, at least the labeling of panels should lie in the same line). The error bars (which I presume are present on Figure 1) are not really visible and the figures appear to be low resolution.

Required corrections and improvements were made to Figures 1- 5.

4. PBS is abbreviated but not written in full the first time.

The PBS was defined in full (Page 4, paragraph 3).

5. Reference formatting is inconsistent.

Reference formatting has been corrected.

6. Page 8, line 6: “involved into” ?

This sentence was corrected (Page 8, paragraph 2).

7. There are numerous dashes that seem unnecessary. Page 4 paragraph 3: “label - nucleoli, nuclear speckles...” and page 8: “coilin-EGFP, - a marker protein”. Page 11 paragraph 2: “water, - around 80”.

Dashes were removed.

REVIEWERS' COMMENTS:

Reviewer #1 (Remarks to the Author):

I think the authors have addressed the issues raised and I am happy for this manuscript to be published in its current form.

While I believe that variations in pH and temperature can impact fluorescence lifetimes of some fluorescent proteins (e.g. CFP), I accept that the lifetime of GFP is more stable.

Reviewer #2 (Remarks to the Author):

The authors appear to have done a thorough job clarifying their approach and satisfying our concerns, with additional experiments, references and clarifications in the manuscript. They defined the experimental error in the instrument response and this was indeed significantly smaller than their measured responses in the organelles. And the response of the H2B-EGFP is above the error, but less than the large fluctuations in the nuclear organelles, and that is expected. The calibrations in fixed cells make sense now. Overall - the data and immediate interpretations are sound, and the manuscript appears to be suitable for publication.

REVIEWERS' COMMENTS:

Reviewer #1 (Remarks to the Author):

I think the authors have addressed the issues raised and I am happy for this manuscript to be published in its current form.

While I believe that variations in pH and temperature can impact fluorescence lifetimes of some fluorescent proteins (e.g. CFP), I accept that the lifetime of GFP is more stable.

Reviewer #2 (Remarks to the Author):

The authors appear to have done a thorough job clarifying their approach and satisfying our concerns, with additional experiments, references and clarifications in the manuscript. They defined the experimental error in the instrument response and this was indeed significantly smaller than their measured responses in the organelles. And the response of the H2B-EGFP is above the error, but less than the large fluctuations in the nuclear organelles, and that is expected. The calibrations in fixed cells make sense now. Overall - the data and immediate interpretations are sound, and the manuscript appears to be suitable for publication.

We thank the Reviewers for valuable comments and acceptance of our manuscript. Their thorough reviews helped us to improve the manuscript significantly.